



1  Spatial estimation of soil carbon, nitrogen and phosphorus stoichiometry in complex

2  terrains: a case study of Schrenk's spruce forest in the Tianshan Mountains

4  Zhonglin Xu[1,2], Yapeng Chang[1], Lu Li[1], Qinghui Luo[1], Zeyuan Xu[1], Xiaofei Li[1],

5  Xuewei Qiao[1], Xinyi Xu[1], Xinni Song[1], Yao Wang[3]

[1] College of Resource and Environmental Science, Xinjiang University, Urumqi

830002, China

[2] Key Laboratory of Oasis Ecology of the Ministry of Education, Xinjiang University,

Urumqi 830002, China

[3] Institute of Desert Meteorology, CMA, Urumqi, Urumqi 830002, China

Abstract: Spatial patterns of soil carbon (C), nitrogen (N) and phosphorus (P) and

their stoichiometric characteristics (C:N:P) play an important role in nutrient

limitations, community dynamics, nutrient use efficiency and biogeochemical cycles,

etc. To date, the spatial distributions of soil organic C at various spatial scales have

been extensively studied, whereas little is known about the spatial patterns of N and P

and C:N:P ratios in various landscapes, especially across complex terrains. To fill this

gap, we estimated the spatial patterns of concentrations of C, N and P and C:N:P

ratios in Schrenk's spruce (*Picea schrenkiana*) forest in the Tianshan Mountains using

multiple linear regression (MLR) model based on data from soil profiles collected

from 2012 to 2017. We found that (1) elevation and climatic variables jointly

contributed to concentrations of C, N and P and C:N:P ratios, (2) soil concentrations

and stoichiometric ratios demonstrated different but continual spatial patterns in

Schrenk's spruce forest, and (3) MLR models could be reliably used to estimate the

spatial patterns of soil elemental concentrations and stoichiometric ratios in

mountainous terrain. We suggest that more independent variables (including biotic,

abiotic and anthropogenic factors) should be considered in future works. Additionally,

adjustment of MLR and other models should be used for a better delineation of spatial

patterns in the concentrations of soil elements and stoichiometric ratios.





Keywords: Spatial distribution, element concentrations, C:N:P ratios, *Picea*

*schrenkiana*, Tianshan Mountains

Introduction

Like those in grasslands and croplands, stocks of macro-nutrients such as carbon (C),

nitrogen (N) and phosphorus (P) in subalpine forests play an important role in the

terrestrial biogeochemical cycle due to their large stocks and potential feedbacks to

external disturbances (Allison and Treseder, 2010; Huang and Schoenau, 1996; Cao

and Chen, 2017). Theoretically, C, N and P are proportionally organized in

ecosystems and controlled by plant functional type, climate and human activities (De

Long et al., 2016; Tian et al., 2010b; Vitousek, 2002). Since the relative stocks of

these nutrients (as well as other nutrients such as potassium, calcium, sodium and

magnesium) are crucial for the accumulation and allocation of plant biomass under

various environmental stress such as drought, heat and light (Lie and Li, 2016;

Niinemets, 2010), the science of ecological stoichiometry, which focuses on nutrient

ratios and the impact factors of these ratios, has been proposed as a tool to study the

spatial-temporal variation and driving forces of ratios across different ecosystems

(Elser et al., 2000; Sterner and Elser, 2002; Mcgroddy et al., 2004). Currently,

ecological stoichiometry has been successfully used to study nutrient limitation

(Feller et al., 2003; Högberg et al., 2017), community dynamics (Johnson and

Agrawal, 2005), microorganism nutrient status (Hill et al., 2012), symbiosis

relationships (Mariotte et al., 2017), nutrient use efficiency (He et al., 2010) and the

global biogeochemical cycle (Schmidt et al., 2016; Midgley and Phillips, 2016) in

terrestrial and aquatic ecosystems.

Soil is an important component of global biogeochemical cycles. Spatial patterns of

soil C, N and P stocks and stoichiometry can contribute as input of independent

validation for global and biogeochemical models and consequently provide valuable

information to examine the feedbacks of different terrestrial ecosystems to global



environmental change (Yang et al., 2010). However, previous studies conducted in

terrestrial ecosystems focused largely on vertical patterns of C, N and P and the

stoichiometric ratios resulting from outputs due to plant growth and inputs from litter

decomposition and soil weathering (Jobbágy, 2000; Yang et al., 2010; Wang et al.,

2015), and the elevation patterns induced by climatic gradient and disturbances

(Müller et al., 2017; Richardson, 2004). To date, the spatial distribution of these

nutrients, and especially their stoichiometric ratios, remains poorly understood. Since

the driving forces that play a part in vertical and elevational variations of nutrient

stocks and stoichiometric ratios can also play a role across landscapes and ecosystems,

an improved understanding of spatial distribution of soil C, N, P and C:N:P ratios and

their driving forces is urgently needed to spatially quantify the responses and

feedbacks of nutrient stocks and stoichiometric ratios to spatial variation in plant

uptake, biomass accumulation and other ecological processes.

Recently, Liu et al. (2013) studied the spatial patterns of soil total N and P across the

Loess Plateau region of China and found that concentrations of N and P are relatively

higher in forestland with higher precipitation and temperature. In addition, their

results also revealed that soil N and P in forestland demonstrated moderate spatial

dependence (Liu et al., 2013). In alpine treeline ecotones, Müller et al. (2017) studied

the availability of soil organic matter, N and P and found that P decrease significantly

as elevation increases. Moreover, the type of tree species mainly contributed to spatial

discrepancies in soil nutrient status (Müller et al., 2017). Compared to these forest

ecosystems, subalpine forests have unique climatic, topographic and pedological

characteristics that closely relate to soil nutrient status and thus may result in different

soil nutrient patterns. However, limited information is available regarding the spatial

pattern of soil C, N, and P in subalpine forests. Therefore, systematic investigation of

soil C, N, and P in subalpine forests is urgently needed in order to improve our

understanding of spatial variations of the nutrient cycle in forest ecosystems and their

responses to global environmental change.



In the past several decades, soil C:N:P ratios have been extensively studied due to the

close relationship between C, N and P and the role of the ratio as an indicator of

nutrient limitations on plant growth. Like the limited references to spatial distribution

or variation in soil C, N and P, available information on spatial patterns of soil C:N:P

is also rare. Recently, a study that aimed to explore general soil C:N:P ratios and the

patterns in these ratios with regard to soil depth, developmental stages and climate

was conducted in China and revealed that while C:N ratios showed relatively small

variation among climatic zones, C:P and N:P showed high spatial heterogeneity and

large variation between different climatic zones (Tian et al., 2010a). In addition,

Sardans et al. (2016) examined the soil concentrations of N and P, as well as N:P

ratios, and discussed the relationship between the ratio and climate across European

*Pinus sylvestris* forests. According to their results, the soil N:P ratios in these forests

displayed spatial variation due to the limiting roles of P and higher levels of N

deposition in the center of the species' distribution. Although these two studies and

other investigations conducted in different ecosystems shed some light on the spatial

distribution of stoichiometric ratios, they only considered the trend of the ratios in a

transect comprising a series of sampling sites distributed along latitudinal, elevational

or climatic gradients, rather than a spatial distribution or pattern of ratios that had the

potential to characterize relative nutrients supply across the soil surface or landscape

of terrestrial ecosystems. Inspired by species distribution models, the pioneer work

conducted by Leroux et al. (2017) developed stoichiometric distribution models to

map the spatial structure of nutrient composition across a landscape and evaluate the

spatial responses of consumers to the composition. They argued that spatial patterns in

nutrient composition may uncover ecosystem properties that are not revealed by

approaches routinely used in ecological stoichiometry. Thus, an improved

understanding of the spatial distribution of C:N:P stoichiometric ratios in different

ecosystems is urgently needed in order to predict and understand the effects of

changing biogeochemistry on ecosystem function and services(Leroux et al., 2017).

The Schrenk's spruce (*Picea schrenkiana*) forest under study is a typical subalpine



forest in the Tianshan Mountains. The forest ranges from Uzbekistan to the mountains

of northwestern China, spanning more than 1800 km in longitude. The elevational

range of the species is 1600-2800 m a.s.l. The broad elevational range and unique

location of the species offers a unique opportunity to study the spatial distribution of

soil C, N, P and C:N:P stoichiometric ratios in subalpine forest. Using a dataset of

several sampling sites across the distribution of Schrenk's spruce forest, a previous

study conducted by Chen et al. (2008) reported that the soil nutrient stocks in eastern

distribution of Schrenk's spruce forest were relatively poor compared to other areas

due to the variation in mean annual precipitation (MAP). In addition, Dai et al. (2013)

studied the spatial variation of treeline and soil nutrient characteristics in a Schrenk's

spruce forest and found that the C stock was relatively higher in central sites, while

total N and P were relatively higher in the western area. To date, limited data are

available regarding the spatial distribution and variation of soil C, N, P and C:N:P

stoichiometric ratios in the Schrenk's spruce forest. Since the soil nutrients and

stoichiometric ratios are closely related to spatial variations in climate

conditions(Yang et al., 2010; Sardans et al., 2016; Müller et al., 2017) and the climate

continuously varied from the western to eastern parts of the Tianshan Mountains

(Wang et al., 2011), a continued spatial pattern of soil C, N, P and C:N:P

stoichiometric ratios could be expected.

In this study, we first investigated soil C, N, P and C:N:P ratios in Schrenk's spruce

forest. The data were obtained from field investigation and laboratory analysis. We

then conducted statistical analysis in order to construct the multiple linear regression

(MLR) relationship between the independent variables (soil C, N, P and C:N:P ratios)

and dependent variables (elevation, MAT, MAP, etc.). By applying the relationship

across the distribution of the Schrenk's spruce forest, we examined the spatial pattern

of soil C, N, P and C:N:P ratios in Schrenk's spruce forest. We hypothesized that (1)

soil C, N, P and C:N:P ratios could be delineated using a linear combination of

elevation and climatic variables and that (2) the spatial distribution of soil C, N, P and

C:N:P ratios would demonstrate a continuous pattern in the Schrenk's spruce forest.



## 2. Materials and methods

### 2.1 Study area

The Tianshan Mountains lie in Central Asia. Our study area (Figure 1) is located in the central and eastern parts of the mountains (N 42°35' - 44°20', E 80°14' - 88°07') and is characterized by a continental climate, with a cold and dry winter and a warm and humid summer. Due to the unique geographic location and topographic characteristics, the daily temperature range is higher than in surrounding regions. Mean annual sunshine duration is more than 2000 hours. MAT decreases from 13.3 °C at lower elevations to −7.3 °C at higher ones, while MAP increases from less than 100 mm to over 800 mm with increasing elevation (Li et al., 2016). The maximum temperature of the warmest month, minimum temperature of the coldest month, and precipitation of the wettest quarter range from 10.6 to 30.4 °C, from -29.2 to -17.1 °C and from 36 to 104 mm in the Schrenk's spruce forest of the Tianshan Mountains. Generally, soil in the forest begins to freeze in late November and starts to melt in early April. Spring and autumn are short due to a relatively longer summer and winter. The vegetation types in the Tianshan Mountains include (from low to high elevations) steppe, steppe–forest, subalpine shrubby meadow, alpine–frostaction–barren zone, and permanent snow and ice (Li et al., 2016). Schrenk's spruce forests form single-species stands between 1,600 m a.s.l. and 2,700 m a.s.l. Shrub species growing at the forest margins include *Cotoneaster melanocarpus, Berberis heteropoda, Rosa spinosissima, Spiraea hypericifolia, Juniperus pseudosabina, Caragana leucophloea* and *Lonicera hispida.* Understory herbal species include *Geranium rotundifolium, Alchemilla tianschanica* and *Aegopodium podagraria*(Wang et al., 2016).



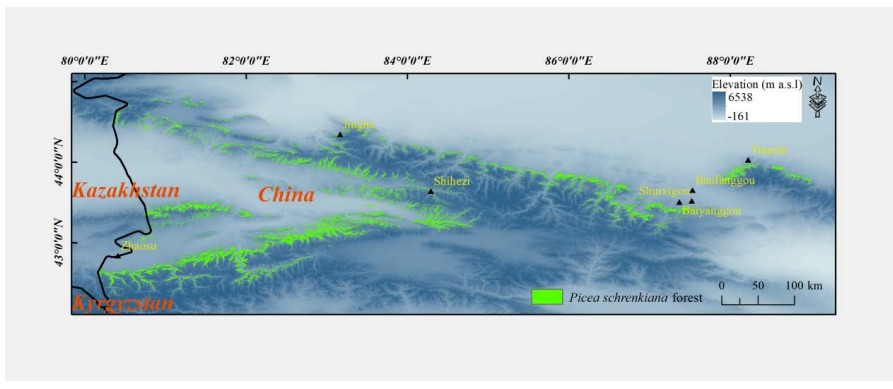

Figure 1. Location of the study area (Tianshan Mountains) and sampling sites (black

triangles).

2.2 Field sampling and laboratory analysis

Fieldwork was conducted from 2012 to 2017. We selected seven sites (Zhaosu, Jinghe,

Shihezi, Baiyanggou, Banfanggou, Shuixigou and Tianchi as shown in Figure 1) in

the Schrenk's spruce forest in order to sample the soils. At each site, several transects

(slopes with continuous elevations and with a distribution of Schrenk's spruce forest)

were selected, and along the elevational gradient of these transects, a certain number

of sampling plots were set up at approximately 30 m elevational intervals. The

longitude, latitude and elevation of each plot were recorded with a global positioning

system receiver, i.e., eTrex venture with 5 m precision. Soil samples at 10 cm

intervals were collected using a soil auger from three random parallel profiles within

each plot. The collected parallel samples were mixed to obtain a composite sample for

each depth. After being taken to the laboratory, the soil samples were air-dried and

sieved for the subsequent chemical analysis. The soil carbon concentration was

measured using the potassium dichromate method as demonstrated in Yeomans and

Bremner (1988). Total N concentration ($g\ kg^{-1}$) was analyzed using the Kjeldahl

digestion method (Bremner and Tabatabai, 1972). Total P concentration ($g\ kg^{-1}$) was

measured using the perchloric acid digestion method, followed by the molybdate

colorimetric test (Sherman, 1942). Soil C, N and P were expressed as $g\ kg^{-1}$ based on

dry weight. Stoichiometric data were calculated by mass ratio. Prior to further



analysis, we checked the normality of collected data using Kolmogorov-Smirnov test.

2.3 Climate data

Climate data were obtained from the Worldclim (version 1.4) bioclimatic dataset

(Hijmans et al., 2005), which provides data as 50-year means for averages and

extremes of precipgitation and temperature. Based on data observations from more than

40,000 weather stations, climatic variables in the dataset were interpolated using a

thin-plate smoothing spline algorithm and latitude, longitude, and elevation as

independent variables (Hijmans et al., 2005). The data have been successfully used

for various spatial modeling applications(Fick and Hijmans, 2017). The original

spatial resolution of the variables in the dataset was 1km×1km, and further processing

in order to extract the required variables resulted in a resolution of 30m×30m. When

determining the response of soil C, N, P and C:N:P ratio to climatic variables, we

selected MAT, MAP, mean temperature of wettest quarter (TWT), mean temperature

of warmest quarter (TWM) and precipitation of warmest quarter (PWQ) in the dataset

as the independent variables. These three variables (TWT, TWM, PWQ) were selected

due to their direct and indirect controls over biomass production, soil microbial

decomposition and nutrient stocks in terrestrial ecosystems (Peri et al., 2015;

Deblauwe and Murray, 2008).

2.4 Actual distribution of the Schrenk's spruce forest and spatial estimation of C, N, P

concentration and C:N:P ratio

The actual distribution of the Schrenk's spruce forest is necessary for spatial

estimation of C, N, and P concentrations and the C:N:P ratio in the forest. In this study,

the actual distribution of the Schrenk's spruce forest was compiled by Hou et al.

(2001) and obtained from the Environmental and Ecological Science Data Center for

West    China,    National    Natural    Science    Foundation    of    China

(http://westdc.westgis.ac.cn). The original vector data were first converted to raster

format using conversion tool and subsequently resampled to 30m×30m resolution

using raster processing tools in ArcToolbox of ArcMap 10.0 (ESRI Inc). The spatial



distributions of soil C, N and P concentrations and C:N:P were calculated using the

constructed MLR models and extracted elevation and climatic variables (MAT, MAP,

etc.) at pixels within the Schrenk's spruce forest. Specifically, the models is

$$Y = \beta_0 + \beta_1 X_1 + \beta_2 X_2 + ... + \beta_k X_k + \varepsilon \tag{1}$$

where $Y$ is the dependent variables (C, N, and P concentrations and C:N:P ratios),

$\beta$s are the regression coefficients of independent variables $X$s, and $\varepsilon$ is the error.

The performances of the MLR models were evaluated using mean absolute

percentage error by

$$M = \frac{100}{n} \sum_{t=1}^{n} \left| \frac{A_t - F_t}{A_t} \right| \tag{2}$$

where $A_t$ denotes the actual value and $F_t$ is the modeled value. The determination of

elevation and climatic variables for C, N P concentration and the C:N:P ratio in the

MLR function were evaluated using the coefficient of determination ($R^2$). Correlation

and regression relationships were considered significant if the calculated value was

greater than the threshold value at a 0.05 significance level. All the above mentioned

statistical analyses were carried out using Origin 8.5 (OriginLab corporation). The

Origin software was also used to prepare the figures.

3. Results

3.1 Regression models for soil C, N and P concentrations and C:N:P ratios

The histograms of C, N and P concentrations and C:N, C:P and N:P ratios are

displayed in Fig. 2. The Kolmogorov-Smirnov normality test demonstrated that the

collected data (C, N and P concentrations and C:N, C:P and N:P ratios) were

significantly drawn from normally distributed populations (0.05 confidence level).

MLR model revealed that elevation positively contributed to C and N concentration

and N:P ratios in the Schrenk's spruce forest, whereas P concentration, C:N and C:P

responded negatively to elevation (Table 1). MAT and MAP both had a positive effect

on C concentration and C:P ratios; however, their increase may be accompanied by a

decrease in C:N and N:P ratios. C, N and P concentrations and C:N ratios decreased



with an increase in TTQ; in contrast, C:P and N:P may increase with increasing TTQ

(Table 1). The increase in TWM had a positive effect on N and P concentrations and

C:N and N:P ratios. C, N and P concentrations responded positively to PWQ, whereas

their ratios responded negatively to the variable (Table 1). The MLR model of P and C

concentration had the highest and lowest coefficients of determination ($R^2$): 0.42 and

0.16, respectively. All regression models were significant at a 0.05 confidence level,

and among the six models, the N and P concentration and C:N and N:P models were

significant at 0.01 confidence level.

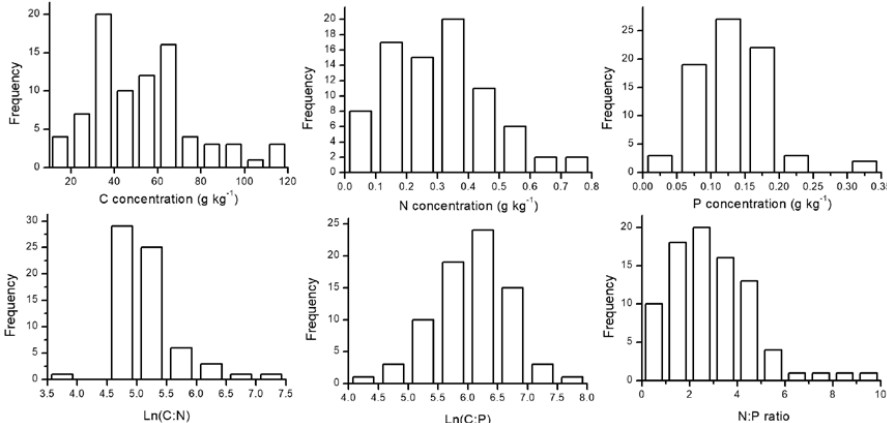

Figure 2. Frequency distribution of soil C, N, P, Ln(C:N), Ln(C:P) and N:P in our

study area. Sample sizes for C, N, P, Ln(C:N), Ln(C:P) and N:P were 83, 81, 76, 66,

76 and 85, respectively. According to results of the Kolmogorov-Smirnov test, all the

data were significantly drawn from a normally distributed population ($p<0.05$).

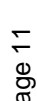



Table 1. Summary of multiple linear regressions of soil C, N, P, Ln(C:N), Ln(C:P) and N:P.

| Independent variables | Regression coefficients of dependent variables | | | | | | Interception | N | $R^2$ | F |
|---|---|---|---|---|---|---|---|---|---|---|
| | Elevation | MAT | MAP | TTQ | TWM | PWQ | | | | |
| C concentration | 0.01 | 3.90 | 0.16 | -6.42 | -1.19 | -0.72 | 204.44 | 83 | 0.16 | 2.44* |
| N concentration | 1.59E-4 | -0.02 | 2.15E-3 | -0.04 | 6.75E-3 | -9.53E-3 | 1.29 | 81 | 0.37 | 7.27** |
| P concentration | -9.18E-5 | 0.02 | -1.21E-3 | -0.09 | 0.04 | -7.05E-4 | 1.47 | 76 | 0.42 | 8.45** |
| Ln(C:N) | -1.69E-3 | -0.04 | -9.32E-3 | -0.75 | 0.69 | 0.01 | 9.74 | 66 | 0.33 | 4.82** |
| Ln(C:P) | -1.81E-4 | 0.02 | 1.10E-3 | 0.15 | -0.06 | 7.75E-3 | 3.65 | 76 | 0.19 | 2.77* |
| N:P | 5.98E-4 | -0.52 | -1.85E-5 | 0.41 | 0.24 | 4.10E-3 | -7.39 | 85 | 0.28 | 5.13** |

Notes: * indicates $p < 0.05$, ** indicates $p < 0.01$. MAT: mean annual temperature; MAP, mean annual precipitation; TTQ, mean temperature of wettest quarter; TWM, mean temperature of warmest quarter; PWQ: precipitation of wettest quarter.



3.2 Spatial patterns of soil C, N and P concentrations and C:N:P ratios

Measured C concentrations ranged from 15.85 to 115.67 g kg$^{-1}$ with an average of 52.81 g kg$^{-1}$. The modeled range is 24.72 - 83.96 g kg$^{-1}$ with an average of 53.15 g kg$^{-1}$ (Table 2). Standard deviation (SD) of the modeled value was lower than that of the measured ones. The range of modeled N concentration (0.11 - 0.53 g kg$^{-1}$) was relatively narrower than for the measured values. Similar to SD of the C concentration, the modeled N concentration also had a lower SD than the measured value. For P concentration, a relatively wider range was found in modeled values (0.04 - 0.44 g kg$^{-1}$) than measured values (0.04 - 0.34 g kg$^{-1}$), which also resulted in a relatively higher average concentration (0.22 g kg$^{-1}$ compared to 0.13 g kg$^{-1}$); the SD of the modeled P concentration was higher than that of the measured value. The average value and SD of the modeled C:N ratio (180.53 and 123.84) were lower than the measured average and SD (214.16 and 200.28). C:P and N:P ratios in the modeled results have narrower ranges than those of the measured values. In addition, the modeled averages of the two ratios were higher than those of the measured ones. As in the other estimations, the SDs of these two ratios in the modeled results were lower than those of the measured data (Table 2).

Table 2. Comparison between measured and modeled C, N, P, Ln(C:N), Ln(C:P) and N:P.

|  | Measured | | | Modeled | | |
|---|---|---|---|---|---|---|
|  | Range | Average | SD | Range | Average | SD |
| C concentration | 15.85-115.67 | 52.81 | 23.00 | 24.72-83.96 | 53.15 | 7.56 |
| N concentration | 0.02-0.74 | 0.31 | 0.16 | 0.11-0.53 | 0.36 | 0.07 |
| P concentration | 0.04-0.34 | 0.13 | 0.06 | 0.04-0.44 | 0.22 | 0.07 |
| C:N ratio | 49.43-1327.37 | 214.16 | 200.28 | 15.15-1574.52 | 180.53 | 123.84 |
| C:P ratio | 80.56-2339.14 | 514.57 | 350.98 | 138.99-2064.22 | 538.99 | 281.33 |
| N:P ratio | 0.22-9.74 | 2.96 | 1.76 | 2.00-4.03 | 3.04 | 0.37 |

The measured concentrations and ratios were values at sampling sites. The modeled values were extracted from spatially interpolated results. SD denotes the standard deviation.



C concentration was highest in the eastern part of the study area, while the central and southwest parts corresponded to relatively lower C concentrations (Fig. 3a). The northwestern study area also had a relatively higher C concentration. N concentration displayed an opposing trend with C concentration: a relatively lower concentration appeared in the eastern study area, while the southwestern part had a relatively higher concentration (Fig. 3b). The trend in the P concentration was similar to that of the C concentration, which increased from west to east in the study area (Fig. 3c). C:N ratios showed a complex pattern in the western part of the study area, whereas generally, the ratios decreased from west to east, as illustrated in Figure 4a. C:P ratios showed a spatial pattern similar to that of the C:N ratios (Fig. 4b). In contrast with the overall trend of C:N and C:P ratios, N:P ratios were lower in the west, higher in the east, and exhibited an increasing trend from the western to eastern parts of the study area (Fig. 4c).





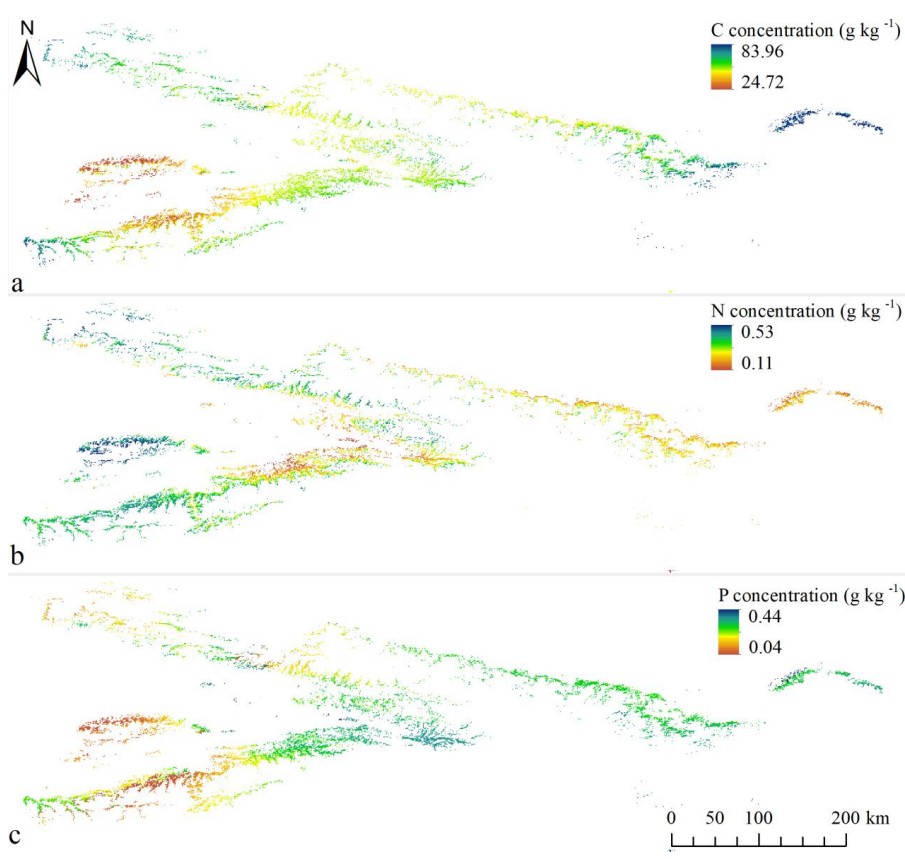

Figure 3. Spatial distribution of soil C, N and P in the *P. schrenkiana* forest.



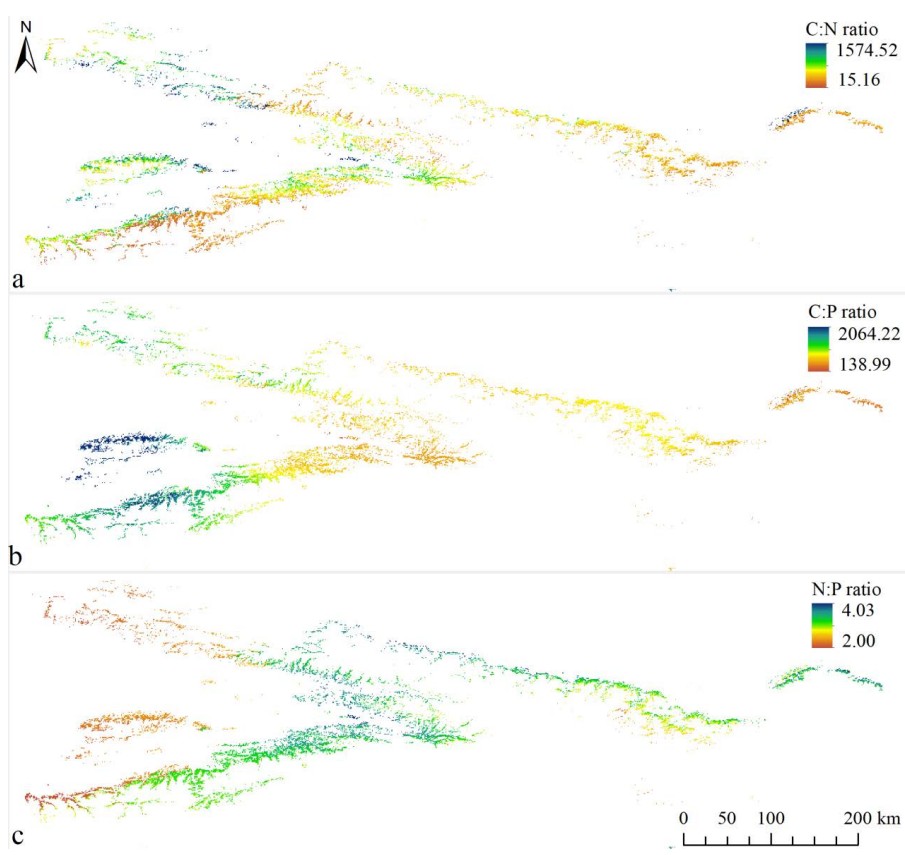

Figure 4. Spatial distribution of soil C:N, C:P and N:P in the *P. schrenkiana* forest.

### 3.3 Comparison between measured and modeled value

The correlation coefficient (*r*) between actual and modeled C concentration was 0.41 (p<0.01, Fig. 5a). The majority of measured C concentrations were higher than 40 g kg$^{-1}$, whereas in Fig. 5a, more scatters with modeled values lower than 40 g kg$^{-1}$ appeared, indicating restricted performance of the multiple regression model. Relatively higher *r* (0.61) with a high confidence level (p<0.01) between measured and modeled N concentrations (Fig. 5b) suggested reliability of the N concentration model compared with the C model. The *r* between measured and modeled values of P concentration was highest among the C, N and P concentrations at 0.75 (p<0.01, Fig. 5c), indicating that the P model was the most reliable among the three concentration models. Fig. 5d displays the scatter of measured and modeled Ln(C:N) at our





sampling sites: a correlation coefficient of 0.57 with 0.01 significance level between measured and modeled Ln(C:N) suggested the reliability of the model, whereas the scatters at the lower-right corner indicated the moderate confidence of the model to the higher values of the C:N ratios. Moderate and significant $r$ values (0.51 and p<0.01) between measured and modeled values of Ln(C:P) were partly impacted by the restricted estimation of the median value of the C:P ratios, as displayed in Fig. 5e. The performance of the N:P ratios model was also acceptable, as shown in Fig. 5f.

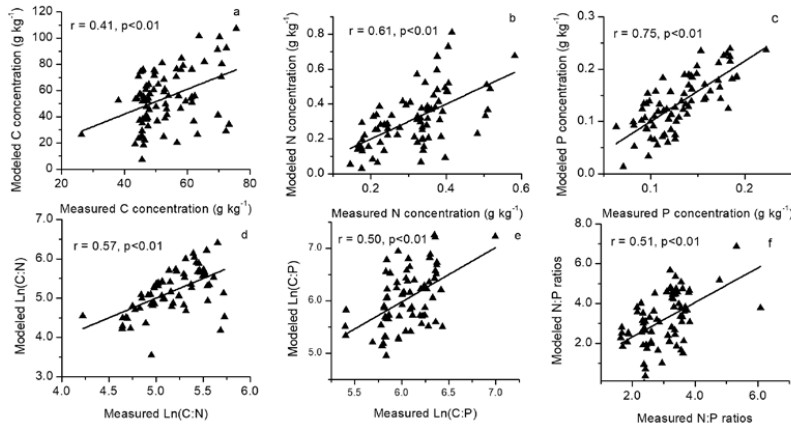

Figure 5. Scatter plots of MLR, estimated and measured C, N and P concentrations, and stoichiometric ratios at the sampling plots. C:N and C:P ratios were log-transformed values.

## 4. Discussion

### 4.1 Spatial patterns of C, N and P concentrations and C:N:P stoichiometry in the Schrenk's spruce forest

We hypothesized that the spatial distribution of soil C, N, P and C:N:P ratios would demonstrate a continuous pattern in the Schrenk's spruce forest. As demonstrated in Figs. 3 and 4, C and P concentrations and N:P ratios increased from west to east within the study area, while N concentration and the other two ratios (C:N and C: P) decreased along the west-east direction. A previous investigation by Dai et al. (2013)revealed similar variations in soil organic C and total N at sampling sites



distributed from west to east in the Tianshan Mountains. According to their results, the differences in soil nutrient concentrations were caused by variation in abiotic variables including temperature, precipitation, evaporation capacity and solar radiation. In addition, a systematic soil survey conducted in the Tianshan Mountains reported relatively lower concentrations of soil C and P in the eastern part of the study area compared with the western part (Cui et al., 1996). It should be noted that the comparison of the concentrations and stochiometric ratios of previous studies were conducted among sampling sites. In contrast, our study not only confirmed the general variations in the concentrations of elements and their stochiometric ratios but also provided the spatial patterns or variations of the concentrations and stochiometric ratios based on the data derived from sampling sites. In fact, these spatial patterns are more useful for a better understanding of the regional biological and ecological processes corresponding to soil chemical characteristics (Prater et al., 2017; Leroux et al., 2017).

## 4.2 Reliability of MLR models

Ecological stoichiometry plays an important role in nutrient limitation (Feller et al., 2003; Högberg et al., 2017), nutrient use efficiency (He et al., 2010), community dynamics (Johnson and Agrawal, 2005), symbiosis relationship(Mariotte et al., 2017) and regional and global biogeochemical cycles (Schmidt et al., 2016; Midgley and Phillips, 2016). However, very few studies have investigated the spatial patterns of variability in nutrient stoichiometry and the implications of these patterns on ecosystem functioning. In their pioneering work, Leroux et al. (2017) reported that the spatial patterning of trait variation within populations is crucial for our understanding of ecological interactions. Therefore, an improved understanding of the spatial patterns of nutrients and their stoichiometric characteristics may contribute to our understanding of material stocks and fluxes in aquatic and terrestrial ecosystems. To date, some studies have been conducted to describe the latitudinal and elevational trends in elemental stoichiometry and their climatic drivers. For example, through analysis of foliar and below-ground biomass samples obtained along a Chinese



grassland transect, Yu et al. (2017) investigated the N:P stoichiometric characteristics of below-ground biomass and foliar and their climatic and altitudinal correlates. According to their results, foliar N and P increased with elevation and below-ground biomass N and P decreased with elevation (Yu et al., 2017). In addition, foliar N decreased and below-ground biomass N increased with MAT (Yu et al., 2017). Sardans et al. (2016) studied foliar and soil concentrations and stoichiometry of N and P across European *Pinus sylvestris* forests and found that Log (N:P) linearly decreased with latitude.

The abovementioned studies shed light on the spatial patterns of nutrients and their stoichiometric ratios in different ecosystems, whereas the corresponding patterns were primarily derived from basic statistical analysis. Specifically, the relationships between concentrations of nutrients and stoichiometric ratios with topographic and climatic variables were obtained based on systematic sampling, Pearson's correlation and simple linear regression analysis (Feng et al., 2017; Agren et al., 2012; Zinke and Stangenberger, 2000). These relationships were useful when examining the response of dependent variables (concentrations and stoichiometric ratios) to independent variables (elevation, latitude, MAT, MAP, etc.). However, their application might be restricted since the spatial variability in the concentrations and stoichiometric ratios across the landscape could not be derived through use of only Pearson's *r* and simple linear regression relationships. Fortunately, some attempts that aim to derive the spatial patterns of stoichiometry have been conducted, and valuable results were reported. These methods include geostatistics and spatial interpolation as reported in Liu et al. (2013), in which the authors obtained a spatial distribution map of soil total N and P density across the Loess Plateau of China. Similar work was also reported elsewhere (Wang et al., 2009; Smith et al., 2014). Unfortunately, the successful application of spatial interpolation might be restricted due to the large amount of samples required for accurate spatial estimation, especially in regions with complex topography. In addition, the specific response of concentrations and stoichiometric ratios to determinants cannot be identified. Remote sensing methods can also be



applied for spatial estimation of ecological stoichiometry. For example, Asner et al. (2015) developed maps of canopy traits, including elemental concentrations and related stoichiometric ratios, in the Amazonian lowland using airborne laser-guided imaging spectroscopy. According to their results, remote sensing technologies have the potential to provide spatially explicit distributional data on multiple canopy foliar traits (including elemental concentrations and stoichiometric ratios) which are unachievable in field studies (Asner et al., 2015). However, as in spatial interpolation methodology, the remote sensing estimates cannot offer the specific response of concentrations and stoichiometric ratios to determinants. Moreover, the traits of soils beneath the canopy cannot be reliably estimated using remote sensing technologies.

Recently, stoichiometric distribution models (StDMs), which imitate the species distribution model and aim to delineate the spatial pattern of elemental stoichiometry, were developed and used to map spatial structures in resource elemental composition and the response of consumers across a landscape (Leroux et al., 2017). Similarly to our approach, the StDMs also fitted generalized linear regression models between elemental response variables and covariates. According to their results, StDMs will allow researchers to map element resources across geographic spaces and hold the promise of describing geographical patterns in organismal elemental traits at various spatial extents (Leroux et al., 2017). In this study, we found that the $R^2$ of the MLR models are high (Table 1), further supporting the reliability of MLR models in spatial estimation of elemental concentrations and stoichiometric ratios. In addition, Rial et al. (2016) also reported good predictive performance of MLR when estimatingsoil C concentration in NW Spain.

### 4.2 Predictor selection

There are three basic categories of variables that could influence soil element concentration and stoichiometric characteristics, namely, the biotic, abiotic and anthropogenic variables (Peñuelas et al., 2015). Biotic interactions, which couple cycles of elements, are therefore critical to spatial delineation of ecological





stoichiometry in terrestrial ecosystems. Currently, the influences of plant uptake on soil nutrient status and elemental stoichiometry have been confirmed in some terrestrial and aquatic ecosystems. For example, See et al. (2015) found that the resorption efficiency of P by plant species increased with soil N content, suggesting that soil nutrient status is correlated with plant uptake. Similarly, in a study focused on linkages of plant and soil elemental stoichiometry and their relationships to forest growth in subtropical plantations, Fan et al. (2015) also reported that nutrient concentrations in soil and plants are tightly linked. The variables correlated to soil microbes should also be included in model construction since variations in microbial stoichiometric ratios were primarily associated with changes in the community structure of soil microbes (Chen et al., 2016).

As we have learned so far from past works, soil elemental concentration and stoichiometry may also be limited by temperature, water, light and other abiotic factors. Generally, there are many interacting factors rather than just one controlling abiotic factor. These factors actually contribute to the stocking and flux rate of the elements and consequently elemental concentrations and stoichiometric ratios. For example, soil C concentrations differ primarily with respect to litter fall and decomposition, which in turn vary with temperature, precipitation, light, etc.(Elser et al., 2000; Tian et al., 2010b). Theoretically, temperature together with precipitation (or available water in soil) mainly determines the plant functional type and controls the biomass accumulation of plants in terrestrial ecosystems (De Long et al., 2016). The more suitable the temperature and precipitation condition, the more litter fall accumulation above soils. Climatic variables also have an impact on microbial activities, which are of vital importance for the decomposition of organic matter (Cleveland and Liptzin, 2007; Delgado-Baquerizo et al., 2017). To date, the most used dataset of climatic variables in ecological modeling consists largely of annual trends (e.g., MAT and MAP), seasonality (e.g., annual range of temperature and precipitation) and extreme or limiting conditions (e.g., TTQ, TWM) of temperature and precipitation. Such climatic datasets include NEW01(New et al., 2002), CliMond



(Kriticos et al., 2012), the Worldclim series (Hijmans et al., 2005; Fick and Hijmans, 2017), etc. Since climate variables in these datasets may be linearly correlated with one another (colinearity) due to interpolation methods (Hijmans et al., 2005; Kriticos et al., 2012), the candidate variables used for linear regression need to be selected in order to find the dataset with the lowest colinearity. The effect of light on soil elements and stoichiometry mainly manifests in soil nutrient use by plants under different solar radiation conditions (Urabe et al., 2002). In addition, Martyniuk et al. (2016) found that the biomass, photosynthetic parameters and elemental content of plant species in forest ecosystems exhibited close relationships to light availability. Therefore, the radiation and corresponding variables should be considered as important indirect variables in the regression model used for spatial estimation of elemental concentration and stoichiometry.

The combustion of fossil fuel increased the concentration of $CO_2$ in the atmosphere from less than 300 particles per million (ppm) pre-industrial revolution to more than 400 ppm in 2014, according to Tans and Keeling (2014). Through plant photosynthesis and soil respiration, the increased C in the atmosphere could be sequestrated in soil and resulted in an increase of soil C concentration. Accompanied by C increase, human activities also add N into the biosphere through fossil fuel burning, crop fertilization, and anthropogenic $N_2$ fixation, at a rate of approximately 165-259 M ton N year-1; this is equal to the total amount of N fixed naturally (Peñuelas et al., 2012). The altered pools and cycles of C and N, together with human intervention (such as application of phosphate fertilizer and changing of plant community structure) on soil P pools, created a C:N:P imbalance in global pedosphere (Wang et al., 2014). To our knowledge, human activities have not been used in the spatial estimation of concentrations of nutrients and stoichiometric ratios as independent variables. In consideration of the notable impact of human activities on soil nutrient statues, anthropogenic variables, for example, the global human footprint (Sanderson et al., 2002), should be used for model construction.



4.3 Advantages and limitations of MLR

Here, we document the potential application of MLR in spatial estimation of elemental concentrations and stoichiometry. Through modeling the spatial distribution of concentrations of C, N and P and their stoichiometric ratios in Schrenk's spruce forest, we identified advantages of the MLR model: (1) the responses of concentrations and stoichiometric ratios to independent variables could be reliably quantified using the model, (2) the regression parameters of the model could be easily estimated using data obtained from collected samples, (3) the spatial variations in the concentrations and stoichiometric ratios could be delineated using geographic information systems platforms, and (4) the modeled results were reliable according to corresponding evaluation (Table 2 and Fig. 5).

While MLR is only one of the approaches that enable us for spatial estimation, other models have potential in similar applications, and obvious limitations of MLR models also exist to compare with those of other spatial delineation methods. For example, artificial neural networks (ANN) have the ability to detect complex nonlinear relationships between dependent and independent variables and possible interactions between predictor variables (Martiny et al., 2013). In multivariate adaptive regression splines (MARS), interactions between variables can be fitted, and rather than fitting a global interaction between a pair of variables, the interactions are specified locally using basis functions (Lek and Guégan, 1999). Classification and regression trees (boosted regression trees, random forests, etc.) are a valuable addition to statistical approaches for the analysis of complex ecological data due to their invariance to transformations of explanatory variables and procedures for handling missing values (De'ath and Fabricius, 2000). Genetic algorithms (GA) are a global optimization method that mimics the action of natural selection to solve complex optimization problems and provide a very efficient method of convergence towards the ideal solution (Hamblin, 2013).

Conclusions



In summary, we estimated the spatial patterns of concentrations of C, N and P and their stoichiometric ratios in Schrenk's spruce forest in the Tianshan Mountains of China. The corresponding results demonstrated the importance of developing reliable methods to delineate the spatial distributions of nutrient concentrations and stoichiometric characteristics. The results have considerable relevance for studies on regional biogeochemical cycles, particularly in complex terrains. We conclude that the concentrations of C, N and P and their stoichiometric ratios in Schrenk's spruce forest could be accurately estimated use MLR methods and that future works can be improved with more independent variables (biotic, abiotic and anthropogenic factors), with adjustment of MLR models, or with implementation of other models (ANN, MARS, GA) that have been successfully adopted for the spatial estimation of soil organic C (Rial et al., 2016; Zhang et al., 2008; Martin et al., 2014).

Author contribution: ZLX designed the experiments and YC, LL, QL, ZX, XL, XQ, XX, XS and YW carried them out. ZLX, LL and YC performed the simulations. ZLX prepared the manuscript with contributions from all co-authors.

Competing interests: The authors declare that they have no conflict of interest.

Acknowledgment

Financial support for this study was provided by the National Science Foundation of China (Nos. 41361098, 31500398, and 31400409). We would like to thank Dr. Abudukeremujiang Zayiti for his assistance during the laboratory analysis.

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
