# Peer review of "Spatial estimation of soil carbon, nitrogen and phosphorus stoichiometry in complex"

_Biogeosciences, 2017_

## Referee Comment (RC1) · Anonymous Referee #1 · 3 May 2018

The authors (Xu et al) have present the results of spatial estimation of soil C, N and P stoichiometry in complex terrains based on a case study of Schrenk's spruce forest in the Tianshan Mountains. The results demonstrated that soil nutrient (C, N and P) concentrations and stoichiometric ratios were related to elevation and climatic variables. This study also discussed the potential application of MLR for estimating their values at large spatial scale. These results are interesting but also are challenged by the methodological questions, and by a lack of clarity in the presentation and interpretation of the study. General and more detailed comments are below. General comments: 1)

[Figure]

The authors stated the application of multiple linear regression (MLR) models in this study. I am not convinced based on the interpretation in introduction and material and method sections. By the way, did you test other models when you stated the reliability of MLR models in discussion? 2) How did you choose the climatic variables (MAT, MAP or others) or did you test these variables in the models based on Aikake Information Criterion (AIC)? Are you sure all the dependent variables (C, N, P and C:N:P ratios) have the same independent variables such as MAT, MAP, Elevation, TTQ, TWM PWQ? 3) I do not understand the purpose of 3.1in result section. It seems to be not well-linked in your result. Could you explain it a little more? 4) There are no page number and line number after line 274. There are a lot of questions can be not listed.

Specific comments: Lines 13-16 This sentence is correct, but not understandable here. Please rewrite and clarify it. Lines 21-22 How many sampling sites do you have collected from 2012-2017? Line 22 which climate variables? Lines 23-25 Please clarify the "different"? Lines 27-28 Suggest to delete it since it is not related to you study. Line 29 Did you analyze the results by other models? Line 30 It may be better to add a sentence to highlight you work and the contribution. Lines 42-49 it is too long and complex. Please rewrite it. Line 49-55 Please write the important ones that are closed to this research. Line 55 Why add the aquatic ecosystems, but this is not your focus on. Line 65 which disturbances? Please add them. Line 67-73 It is not long, and not understandable. Line 75 The year of 2013 is not recent? Line 78-79 What is the meaning for moderate spatial dependence? Lines 104-110 Please rewrite this sentence. Lines 147-149 Why only combination of elevation and climate variables? Why it is a linear regression but not nonlinear? Line 176 The size of fig 1 is too small. Please add more information on the sampling sites and total numbers. Please clarify why these sites can be representative for the whole forests. Lines 211-217 Did you test the variables based on your data? Lines 229-234 as proposed in general comment 2. There is no line number after line 274. The fig 3, 4 are not high quality. Please change it. 4.2 How can you conclude that reliability of MLR models since you did not analyze the data using other models. There are repeat literature in the references. Please

revise it.

revise it.

Please also note the supplement to this comment:
https://www.biogeosciences-discuss.net/bg-2017-536/bg-2017-536-RC1-supplement.pdf

---

## Referee Comment (RC2) · Anonymous Referee #2 · 4 May 2018

Comments on the paper Spatial estimation of soil carbon, nitrogen, and phosphorus stoichiometry in complex terrains: A case study of Schrenks's spruce forest in the Tienshan Mountain submitted to "Biogeosciences Discussions" by Xu et al.

General Comments: The paper of Xu et al addresses an ecologically important issue: The stoichiometry of C, N, and P in forest ecosystems/forest soils is probably a key factor governing growth response patterns as well as community shifts induced by climate change and/or elevated atmospheric $CO_2$ concentrations. Adequate spatial estimation and modelling of ecosystem or soil C:N:P stoichiometry is important to correctly model

and predict such tree and forest growth response patterns on a large spatial scale, particularly in regions where data on soil C, N, and P are sparse.

Despite this general relevance, I unfortunately must suggest rejection of the paper due to several problems with the manuscript, which are described in more detail below. The most serious problems refer to the soil sampling technique and the reported N concentration data, which both are critical for the results reported in the paper.

1. Soil sampling: According to the Materials and Methods Section, soil samples at 10 cm intervals were collected using a soil auger at each study site. However, the topsoil of many forest soils consists of two separate sections which strongly differ with respect to their ecological traits as well as their C:N:S stoichiometry: The forest floor (organic surface layer with an OC concentration > 15 mg/g) and the Ah horizon (humic mineral topsoil with a OC concentration <15 mg/g). I suppose that at least in some of the samples (according to the reported range of C concentrations an the statement on "litter fall accumulation above soils" made by the authors on page 20) a forest floor is present. Pooling forest floor and humic mineral topsoil to a single is not scientifically sound.

2. More important, the reported soil C:N ratios (49-1327) are far too wide to be true. The C/N ratio of microbial organic matter is around 8, that of typical forest topsoils ranges between 15 and 30. The extremely wide C/N ratios presented in the paper probably are caused by wrong (far too small) N concentrations (0.02- 0.74 mg/g). As the N concentrations are a key parameter for the stoichiometry assessments in the paper, the entire paper is based on wrong input data. In my opinion this is a crucial fault making a publication impossible.

3. Probably also the P concentrations have been underestimated, because perchloric acid digestion does not completely mobilize/recover silicate-bound P, resulting in wrong C:P stoichiometry data.

4. Moreover, I want to emphasize that the MLR model partly is based on inappropriate

assumptions. In contrast to the authors I do not think that the reported soil C and N concentrations as well as N:P ratios are normally-distributed, but skewed (as clearly to be recognized in the Histograms presented in Figure 2 and the Scatter plots presented in Figure 5a,b. Probably the K-S test has been applied inadequately.

5. Furthermore, in contrast to the statement made by the authors in the paper, the model explains only a small part of the data variance (according to the $R^2$ values presented in Table 1 only 16% for C, about 40% for N and P). This means that 84% of the variance of the C concentrations remains unexplained by the model.

The English grammar style and spelling is unsatisfactory.

Specific Comments:

The introduction is far too long. It must be shortened considerably. L68 Citation Müller et al 2017: wrong position in reference list. L127 "Soil nutrient stock": Very general and vague phrase. L144/145: Independent and dependent variables probably are mixed up here. L188/189: Why was the soil sampled using an auger, even though profiles were available?

Page 11ff: No line numbers are given.

Results Section: Results are presented "over/pseudo-exact" (not warranted by the analysis precision): e.g. C concentrations 15.85 g/kg; C:N ratio: 123.84, etc.)

Table 1: No units are given. Data are too "pseudo-precise". C/N ratio unrealistic.

Page 20, lower paragraph: Soil C concentrations are also strongly governed by soil texture (protection of OC by clay, Fe,Al oxides, aggregates). The statement that "the more suitable the temperature and precipitation conditions (are), the more litter fall accumulates above soils" is wrong. O layer accumulation is maximal at particularly wet and cool conditions, which cannot be termed particularly suitable.

Please also note the supplement to this comment:

https://www.biogeosciences-discuss.net/bg-2017-536/bg-2017-536-RC2-supplement.pdf

---

## Author Comment (AC1) · 25 May 2018

Dear Editor, Dear reviewers Thank you for your letter and for the reviewers' comments concerning our manuscript entitled "Spatial estimation of soil carbon, nitrogen and phosphorus stoichiometry in complex terrains: a case study of Schrenk's spruce forest in the Tianshan Mountains" . Those comments are all valuable and very helpful for revising and improving our paper, as well as the important guiding significance to our researches. We have studied comments carefully and the responds to the reviewers' comments are as following: General Comments: (1) The authors stated the application

of multiple linear regression (MLR) models in this study. I am not convinced based on the interpretation in introduction and material and method sections. By the way, did you test other models when you stated the reliability of MLR models in discussion? Response: We will revise and expand the Introduction and Material and Method in response to this comment. Additionally, for the spatial estimation, we are collecting more soil samples across the study area at this time, after the laboratory analysis (predicted to be finished at end of June), we will add the results of Kriging interpolation and nonlinear model.

General Comments: (2) How did you choose the climatic variables (MAT, MAP or others) or did you test these variables in the models based on Aikake Information Criterion (AIC)? Are you sure all the dependent variables (C, N, P and C:N:P ratios) have the same independent variables such as MAT, MAP, Elevation, TTQ, TWM PWQ? Response: We agree with the reviewer's worry and stepwise regression based on AIC will be used in the revised manuscript. Actually, we will consider the contribution of other possible variables (climatic variables from Worldclim dataset and topographic variables calculated from DEM) based on PCA analysis, and then the principle components will be used for the spatial estimation.

General Comments: (3) I do not understand the purpose of 3.1in result section. It seems to be not well-linked in your result. Could you explain it a little more? Response: we agree and will modify this section.

General Comments: (4) There are no page number and line number after line 274. Response: we are sorry for the inconvenience and will add page and line number in the revised manuscript.

Specific Comments: Lines 13-16 This sentence is correct, but not understandable here. Please rewrite and clarify it. Response: We agree with the comments and will update accordingly.

Specific Comments: Lines 21-22 How many sampling sites do you have collected from

2012-2017? Response: currently we have seven sampling sites, we are collecting more sites, the specific number of sites will be noted in the revised manuscript.

Specific Comments: Line 22 which climate variables? Response: will add the specific climatic variables in the revised manuscript.

Specific Comments: Lines 23-25 Please clarify the "different"? Response: will delete "different but" in the manuscript.

Specific Comments: Lines 27-28 Suggest to delete it since it is not related to you study. Response: will delete the sentence.

Specific Comments: Line 29 Did you analyze the results by other models? Response: As mentioned in the response to the general comments (1) and (2), we will add more variables and conduct PCA, then use stepwise regression and Kriging interpolation to estimate the spatial pattern of stoichiometric characteristics.

Specific Comments: Line 30 It may be better to add a sentence to highlight you work and the contribution. Response: will modify text.

Specific Comments: Lines 42-49 it is too long and complex. Please rewrite it. Response: we agree and will modify the text.

Specific Comments: Line 49-55 Please write the important ones that are closed to this research. Response: will delete the community dynamics and symbiosis relationships.

Specific Comments: Line 55 Why add the aquatic ecosystems, but this is not your focus on. Response: will delete the aquatic ecosystems.

Specific Comments: Line 65 which disturbances? Please add them. Response: will delete "disturbances".

Specific Comments: Line 67-73 It is not long, and not understandable. Response: We agree with these comments and will update accordingly

Specific Comments: Line 75 The year of 2013 is not recent? Response: will delete "recently" and update text accordingly.

Specific Comments: Line 78-79 What is the meaning for moderate spatial dependence? Response: will delete "moderate".

Specific Comments: Lines 104-110 Please rewrite this sentence. Response: we will rewrite the sentence as suggested.

Specific Comments: Lines 147-149 Why only combination of elevation and climate variables? Why it is a linear regression but not nonlinear? Response: As mentioned in the previous response, we will add more variables and conduct PCA, then use stepwise regression and Kriging interpolation to obtain the spatial distribution. Here, "linear" will be removed and we will find nonlinear relationships.

Specific Comments: Line 176 The size of fig 1 is too small. Please add more information on the sampling sites and total numbers. Please clarify why these sites can be representative for the whole forests. Response: will upload a clear version of Fig.1. Since we are collecting more samples, the specific number of sampling sites will be added in the revised manuscript. The representativeness of these sites will be added accordingly.

Specific Comments: Lines 211-217 Did you test the variables based on your data? Response: will used stepwise regression in the revised manuscript.

Specific Comments: Lines 229-234 as proposed in general comment 2. There is no line number after line 274. The fig 3, 4 are not high quality. Please change it. Response: will add page and line number in the revised manuscript. Fig. 3 and 4 will be replaced by high-quality ones.

Specific Comments: 4.2 How can you conclude that reliability of MLR models since you did not analyze the data using other models. Response: we will add the results of Kriging interpolation and nonlinear models in the revised version.

Specific Comments: There are repeat literature in the references. Please revise it. Response: will deleted the following two repeat literatures: Peñuelas, J., Sardans, J., Rivas-Ubach, A., and Janssens, I. A.: The human-induced imbalance between C, N and P in Earth's life system, Glob. Chang. Biol., 18, 3-6, 2015.

and

Tian, H., Chen, G., Zhang, C., Melillo, J. M., and Hall, C. A. S.: Pattern and variation of C:N:P ratios in China's soils: a synthesis of observational data, Biogeochemistry, 98, 139-151, 2010b.
* * *

---

## Author Comment (AC2) · 25 May 2018

Dear Editor, Dear reviewer We thank reviewer for detailed and constructive comments on the manuscript. Their suggestions have enabled us to improve our work. Based on the instructions provided in your letter, we have studied comments carefully and have made correction which we hope meet with approval. The main corrections in the paper and the responds to the reviewers' comments are as following:

General Comments: Soil sampling: According to the Materials and Methods Section, soil samples at 10 cm intervals were collected using a soil auger at each study site. However, the topsoil of many forest soils consists of two separate sections which strongly differ with respect to their ecological traits as well as their C:N:S stoichiometry: The forest floor (organic surface layer with an OC concentration > 15 mg/g) and the Ah horizon (humic mineral topsoil with a OC concentration <15 mg/g). I suppose that at least in some of the samples (according to the reported range of C concentrations and the statement on "litter fall accumulation above soils" made by the authors on page 20) a forest floor is present. Pooling forest floor and humic mineral topsoil to a single is not scientifically sound. Response: We agree with the reviewer comments about "Pooling forest floor and humic mineral topsoil to a single is not scientifically sound", we will (1) check our soil samples and data, and (2) add more data from soil samples which we are collecting now, the collection will be finished at June and corresponding laboratory analysis will be finished before August. We will update all the tables and figures accordingly.

General Comments: More important, the reported soil C:N ratios (49-1327) are far too wide to be true. The C/N ratio of microbial organic matter is around 8, that of typical forest topsoils ranges between 15 and 30. The extremely wide C/N ratios presented in the paper probably are caused by wrong (far too small) N concentrations (0.02- 0.74 mg/g). As the N concentrations are a key parameter for the stoichiometry assessments in the paper, the entire paper is based on wrong input data. In my opinion this is a crucial fault making a publication impossible. Response: now we are collecting more soil samples across the study area, the collection will be finished at end of June, all the soil samples (samples collected during past fieldwork and during the current one) will be re-analyzed.

General Comments: Probably also the P concentrations have been underestimated, because perchloric acid digestion does not completely mobilize/recover silicate-bound P, resulting in wrong C:P stoichiometry data. Response: again, soil samples will be re-analyzed as soon as possible.

General Comments: Moreover, I want to emphasize that the MLR model partly is based on inappropriate assumptions. In contrast to the authors I do not think that the reported soil C and N concentrations as well as N:P ratios are normally-distributed, but skewed (as clearly to be recognized in the Histograms presented in Figure 2 and the Scatter plots presented in Figure 5a,b. Probably the K-S test has been applied inadequately. Response: now we are collecting more soil samples across the study area, the collection will be finished at end of June, the tables and figures will be updated after we obtain new data.

General Comments: Moreover, in contrast to the statement made by the authors in the paper, the model explains only a small part of the data variance (according to the $R^2$ values presented in Table 1 only 16% for C, about 40% for N and P). This means that 84% of the variance of the C concentrations remains unexplained by the model. Response: We agree with the reviewer's worry and stepwise regression based on AIC will be used in the revised manuscript. Actually, we will consider the contribution of other possible variables (climatic variables from Worldclim dataset and topographic variables calculated from DEM) based on PCA analysis, and then the principle components will be used for the spatial estimation.

Specific comments: The introduction is far too long. It must be shortened considerably. Response: it will be shortened in the revised manuscript.

Specific comments: L68 Citation Müller et al 2017: wrong position in reference list. Response: will modify, thank you.

Specific comments: L127 "Soil nutrient stock": Very general and vague phrase. Response: we checked the paper and found the soil nutrient actually means organic matter, total N and total P, will revise the sentence.

Specific comments: L144/145: Independent and dependent variables probably are mixed up here. Response: we agree and will modify, thank you.

[Figure]

Specific comments: L188/189: Why was the soil sampled using an auger, even though profiles were available? Response: actually, soil profile were hard to collect, we only collected soil samples use auger during our fieldwork, will delete the profile in the revised manuscript.

Specific comments: Page 11ff: No line numbers are given. Response: sorry for inconvenience, we will insert line number in revised manuscript.

Specific comments: Results Section: Results are presented "over/pseudo-exact" (not warranted by the analysis precision): e.g. C concentrations 15.85 g/kg; C:N ratio: 123.84, etc.) Response: soil samples will be re-analyzed, we will update all the data, tables and figures after statistical analysis.

Specific comments: Table 1: No units are given. Data are too "pseudo-precise". C/N ratio unrealistic. Response: as mentioned previously, we will add more data based on current sampling, all tables and figures will be updated.

Specific comments: Page 20, lower paragraph: Soil C concentrations are also strongly governed by soil texture (protection of OC by clay, Fe, Al oxides, aggregates). The statement that "the more suitable the temperature and precipitation conditions (are), the more litter fall accumulates above soils" is wrong. O layer accumulation is maximal at particularly wet and cool conditions, which cannot be termed particularly suitable Response: We agree with the comments and will update accordingly.